# Keypoint-Guided 4D Gaussian Splatting with Decoupled Spatio-temporal Flow Refinement

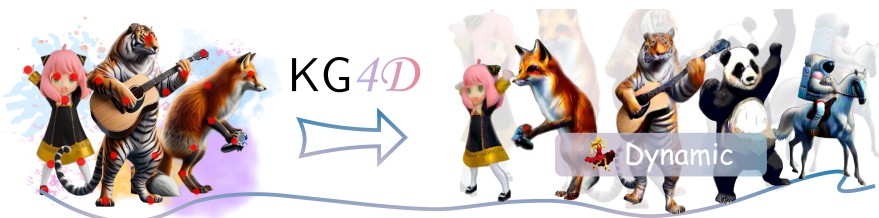

Figure 1: **Visualization of the task:** KG4D receives single images and their corresponding keypoint references (left), and then generates consistent and realistic dynamic 3D models (right).

## Abstract

We propose KG4D, a novel method for generating time-aware 4D representations from a single static image or video. Previous methods largely rely on weak supervision signals, failing to introduce fine-grained supervision necessary for capturing detailed spatio-temporal dynamics. In contrast, our approach employs Harmonic Spatio-temporal Encoding (HSE) to achieve efficient spatio-temporal separation during training, allowing the model to represent dynamic scene changes more accurately. Furthermore, Keypoint Feature Calibration (KFC) ensures precise pose consistency, and Wasserstein Gradient Flow (WGF) enhances motion coherence, effectively reducing artifacts. Comprehensive evaluation and ablations demonstrate that our proposed **KG4D** outperforms existing state-of-the-art methods on various benchmarks in dynamic 4D generation and novel viewpoint synthesis, validating its effectiveness and superior generation capability.

## 1 Introduction

Recent advancements in neural rendering and dynamic modeling (Tewari et al., 2022) with Gaussian splatting (Doe & Smith, 2023) have emerged as a promising approach for generating dynamic scenes by synthesizing 3D content from 2D inputs (Cao et al., 2019), such as images and video sequences. This method leverages probabilistic splatting to create a continuous and temporally coherent representation of dynamic environments (Cao et al., 2019). However, it still faces challenges: (1) Difficulty in maintaining temporal consistency across frames (Chen & Lin, 2023), especially with complex motions and viewpoint changes; (2) Low-dimensional supervision signals struggling to guide the fitting of high-dimensional data distributions and the lack of physically grounded conditioning, leading to unstable and inconsistent outputs. These issues cause instability in capturing fine-grained spatio-temporal local patterns, such as keypoint misalignments and incorrect local topologies in dynamic scenes (Cheng et al., 2020).

To address these challenges, various baseline methods have been proposed. DreamGaussian4D (Zhao & Wang, 2023) introduced 4D Gaussian splatting (Huang et al., 2023a) to improve temporal consistency but struggles with complex motions and viewpoint changes, leading to artifacts and instability due to the inefficiencies in representing dynamic scenes. Other methods, like Consistent4D (Li et al., 2022), Animate124 (Kim et al., 2021), and GaussianFlow (Huang et al., 2023b), focus on pixel-level consistency and geometric constraints but fail to capture the full spatio-temporal dynamics, resulting in artifacts. Similarly, 4Diffusion (Chen & Lin, 2023) enhances spatio-temporal

coherence by multi-view video diffusion (Chen & Lin, 2023) but suffers from frame flickering. Despite progress, these methods remain limited by sparse supervision and the absence of robust physical conditioning.

In this work, we propose KG4D (Keypoint-Guided 4D Gaussian Splatting) to resolve the limitations of current methods effectively. (i) To ensure temporal consistency and reduce artifacts caused by complex motions and varying viewpoints, we introduce Harmonic Spatio-temporal Encoding (HSE), which decouples spatial and temporal dimensions, allowing the model to capture fine-grained sub-structures along different dimensions and achieve smoother transitions across frames. (ii) Building on this, to tackle the problem of insufficient supervision and the inability to capture fine-grained spatial patterns, we propose Keypoint Feature Calibration (KFC). By utilizing keypoint guidance, KFC ensures precise alignment of local topology, achieving pose reconstruction and correcting keypoint misalignments. (3) To further refine the motion dynamics, we leverage Wasserstein Gradient Flow (WGF) for stable, consistent probability flow along temporal subspace, significantly reducing visual artifacts and enhancing the overall quality of motion generation, even in highly dynamic scenes.

We conduct comprehensive evaluations of KG4D across multiple benchmarks, demonstrating an overall 40% improvement among different metrics, particularly in temporal coherence (Mildenhall et al., 2020) and visual quality. However, these improvements come with a 50% reduction in training speed due to the increased computational complexity. Despite this trade-off, KG4D establishes a new benchmark for dynamic 4D scene generation (Chen & Lin, 2023).

Our contributions are as follows:

- We propose a novel Harmonic Spatio-temporal Encoding (HSE) for 4D representation that more effectively captures dynamic scene changes by decoupling spatial and temporal information.

- We introduce Keypoint Feature Calibration (KFC), which provides additional supervision of local patterns in spatial dimensions to ensure accurate keypoint alignment in 3D Gaussian Splatting.

- We implement Wasserstein Gradient Flow (WGF) in the temporal dimension to enhance motion consistency and stability, effectively reducing artifacts and improving visual coherence.

- We establish a new benchmark for 4D Gaussian splatting-based methods through comprehensive evaluations, demonstrating KG4D's state-of-the-art performance in generating realistic, temporally consistent dynamic scenes.

## 2 RELATED WORK

### 2.1 4D GENERATIVE MODELS

4D Gaussian Splatting extends the traditional 3D Gaussian Splatting method by incorporating dynamic changes along the temporal dimension, thereby enabling the modeling of dynamic scenes. Each Gaussian entity possesses attributes such as spatial position, shape, and color, and undergoes corresponding transformations over time. This approach allows for the dynamic capture of motion and changes within the scene while maintaining spatial detail, making it suitable for applications in video content generation and animation production.

### 2.2 CATEGORY-AGNOSTIC POSE ESTIMATION

Keypoint detection plays a central role in pose estimation, motion capture, and dynamic scene generation. It enables the sharing of pose representations across different object categories without the need for independent training for each category. By introducing category-agnostic keypoint representations and a unified feature extraction mechanism, accurate keypoint detection not only captures the pose information of individual entities but also provides essential motion guidance for subsequent three-dimensional dynamic modeling. This method significantly enhances the generalization capability of cross-category pose estimation, offering robust support for pose detection of various objects within dynamic scenes.

## 2.3 Gradient Flows in Wasserstein Metric

The Wasserstein gradient flow (Santambrogio, 2017; Mokrov et al., 2021) refers to the evolution of probability distributions driven by the steepest descent of an energy function with respect to the Wasserstein distance. This flow describes how a probability measure evolves over time by following the gradient of a function in the metric space of probability distributions endowed with the Wasserstein-2 distance. This concept was formalized by (Jordan et al., 1998) (JKO) in 1998 , where they showed that the Fokker-Planck equation can be expressed as a gradient flow in Wasserstein space.

Recent advancements in Gradient flows have since gained significant traction in machine learning and variational inference, applied to problems like density estimation, sampling, and generative modeling, where optimization occurs over distribution spaces (Ansari et al., 2020; Fan et al., 2021). A key commonality between these applications and 4D reconstruction lies in optimizing high-dimensional distributions, where the flow efficiently adjusts distributions in both temporal and spatial dimensions. The connection to optimal transport is central to the Wasserstein gradient flow's desirable properties (Nguyen et al., 2023; Arbel et al., 2019), as the Wasserstein distance originates from minimizing the cost of transporting mass between distributions. This makes the Wasserstein gradient flow particularly suitable for tasks involving dynamic distributions. In our work, we decoupled the 4D Gaussian Splatting process into temporal and spatial dimensions and demonstrated that both satisfy the Fokker-Planck equation, allowing us to leverage the desirable properties (*i.e.* stability, exponential convergence) of Wasserstein gradient flow for efficient gradient-based learning in high-dimensional settings.

## 3 Preliminary

### 3.1 4D Gaussian Splatting

In 3D Gaussian Splatting (3D GS) (Doe & Smith, 2023), a scene is represented by a set of 3D Gaussians $\mathcal{S} = \{G_i\}_{i=1}^N$, each parameterized by its mean $\mu_i \in \mathbb{R}^3$ and covariance matrix $\Sigma_i \in \mathbb{R}^{3\times3}$. The rendered image $I(o)$ from viewpoint $o$ is given by:

$$I(o) = f(G_i, o) = \sum_{i=1}^N T_i \beta_i c_i, \tag{1}$$

$$\text{where } G(x; \mu, \Sigma) = e^{-\frac{1}{2}(x-\mu)^T \Sigma^{-1}(x-\mu)}, \quad \Sigma = RSS^T R^T,$$

$$\beta_i = \alpha_i G_i, \quad T_i = \prod_{j=1}^{i-1}(1 - \beta_j),$$

and $c_i \in \mathbb{R}^H$, $\alpha_i \in [0, 1]$ is the color and opacity of Gaussian $G_i$. To extend this to dynamic scenes, 4D Gaussian Splatting (4D GS) (Huang et al., 2023a) introduces time $t$ as an additional dimension. Each Gaussian $G_{i,t}$ evolves over time via a deformation function $\mathcal{D}$, such that $G'_{i,t} = \mathcal{D}(G_i, t)$. The rendered image at time $t$ becomes:

$$I_t(o) = f(G'_{i,t}, o) = \sum_{i=1}^N T_{i,t} \beta_{i,t} c_{i,t}, \tag{2}$$

where $\alpha_{i,t}$ and $T_{i,t}$ are computed similarly to the 3D case but are now time-dependent. This formulation enables dynamic scene generation by accounting for both spatial and temporal variations.

### 3.2 Probability Flows

In probability flows (PFs), the evolution of a probability density $p(x, \tau)$ over time is governed by the continuity equation:

$$\frac{\partial p(x, \tau)}{\partial \tau} = -\nabla_x \cdot (p(x, \tau)v(x, \tau)), \tag{3}$$

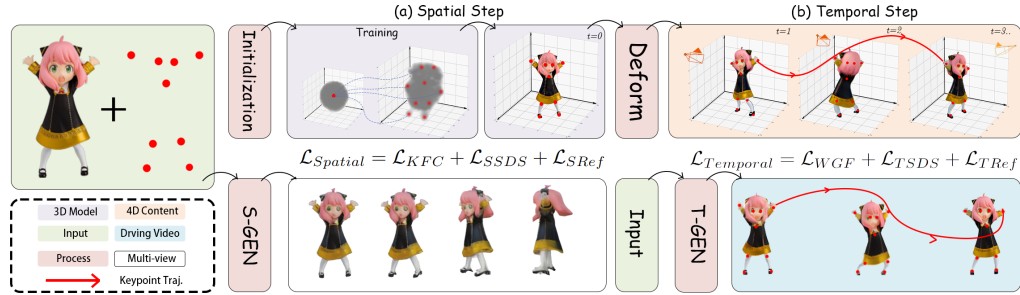

Figure 2: **Overview of KG4D.** We divide 4D probability flow (PF) learning into two stages: spatial and temporal. In the (a) spatial step, we learn 3D PF by fixing $t = 0$ with $\mathcal{L}_{KFC}$ and visual loss $L_{SSDS}$ and $L_{SRef}$ derived from DreamGaussian4D. (b) In the temporal step, we predict keypoint (Zhang et al., 2023b) movements across frames using a deformation network, with Wasserstein offset $\mathcal{H}$ of 2D Gaussians as prediction and optical flow as ground truth (Arbel et al., 2019). In both stages, reference keypoints enforce the model to learn local patterns in 4D Gaussian Splatting.

where $v(x, \tau) = -\nabla_x U(x, \tau)$ is the velocity field derived from a potential $U(x, \tau)$. Alternatively, the state evolution can be represented by an ODE:

$$\frac{dx(\tau)}{d\tau} = v(x, \tau), \tag{4}$$

which traces the trajectory in state space while preserving the probability distribution, allowing for flexible modeling of high-dimensional dynamics. This formalism is particularly suited for modeling the 4D reconstruction process, as it accommodates both spatial and temporal dynamics while allowing the incorporation of supervisory signals to guide the flow in high-dimensional spaces.

## 4 METHODOLOGY

### 4.1 OVERVIEW

**Overall framework.** Learning 4D representations without explicit ground truth is inherently challenging due to the complexity of the high-dimensional data. To tackle this, we introduce KG4D, a two-stage spatio-temporal modeling framework derived from DreamGaussian4D. In the first stage, we optimize a static 3D Gaussian model by leveraging multi-view 2D diffusion priors (Zhang et al., 2023a) to refine pixel-wise geometry and texture, while keypoint (Contributors, 2020) reconstruction and local structural refinement are guided by KFC in 3D space.

In the second stage, a temporal deformation network, denoted as $\mathcal{D}$, is employed to capture dynamic changes via Harmonic Spatio-temporal Encoding (HSE). This network initializes with the 3D Gaussians and keypoint priors derived from the first stage, facilitating further keypoint reconstruction across the temporal domain. Specifically, the 3D Gaussians are embedded using HSE, and $\mathcal{D}$ is responsible for decoding spatial shifts $\Delta\mathcal{X}_t$, rotations $\Delta r_t$, and scaling $\Delta s_t$ at a given time $t$. The HSE embedding is formulated as follows:

$$\text{HSE}(G(x; \boldsymbol{\mu}, \boldsymbol{\Sigma}, t)) = [\sin(\omega_k \cdot [\boldsymbol{\mu}, t]), \cos(\omega_k \cdot [\boldsymbol{\mu}, t])]_{k=1}^{K}$$

where $\omega_k$ represents the frequencies for encoding the spatial and temporal components. The deformation network $\mathcal{D}$ then predict the transformations from these encodings:

$$[\Delta\mathcal{X}_t, \Delta r_t, \Delta s_t] = \mathcal{D}(\text{HSE}(G(x; \boldsymbol{\mu}, \boldsymbol{\Sigma}, t)))$$

This decoupling of spatial and temporal components allows KG4D to perform robust, flexible, and efficient 4D generation.

**Problem formulation.** As shown in Figure. 2, given a support image $I^{sup}$ and its ground truth keypoints $P^{sup}$, the goal is to generate a time-aware 3D Gaussian Splatting (GS) sequence that

represents the dynamic scene based on $I^{sup}$, extending the 2D representation into 4D space to capture both spatial and temporal dynamics. Formally, we treat the 4D Gaussians as Gaussian mixture distributions (GMMs) (Gao et al., 2023) in spatio-temporal space:

$$G(x; \boldsymbol{\mu}, \boldsymbol{\Sigma}) = \sum_{i=1}^{N} \omega_i G_i(x; \boldsymbol{\mu}_i, \boldsymbol{\Sigma}_i) \tag{5}$$

Here, $\boldsymbol{\mu} \in \mathbb{R}^4, \boldsymbol{\Sigma} \in \mathbb{R}^{4,4}$. Therefore, we can model image-to-4D synthesis as a combination of spatial PF and temporal PF, which is proven by Theorem. 1 and 2. This modeling can be satisfied only when the image-to-video mapping is fixed, as such setting provides full-dimensional implicit ground truth for both spatial and temporal domains. Specifically, given initial GMM $G_0(x; \boldsymbol{\mu}_0, \boldsymbol{\Sigma}_0)$ and target GMM $G_{\mathcal{T}}(x; \boldsymbol{\mu}_{\mathcal{T}}, \boldsymbol{\Sigma}_{\mathcal{T}})$, there exist $G_\tau(x; \boldsymbol{\mu}_\tau, \boldsymbol{\Sigma}_\tau), \tau \in [0, \mathcal{T}]$ that satisfy equation 3:

$$\frac{\partial G_\tau(x)}{\partial \tau} = -\nabla_x \cdot (G_\tau(x) v_\tau(x)), \tag{6}$$

where $v_\tau(x)$ denotes the vector field of the PF represented by $G_\tau(x)$. Notably, the 4D ground truth is implicitly encoded within the 2D image plane, and the initial distribution $G_0(x; \boldsymbol{\mu}0, \boldsymbol{\Sigma}0)$ of the PF follows a standard normal distribution. Following this, the goal is to learn a specific function $\Omega\theta(x)$, parameterized by a neural network, for a feasible $v\tau(x)$ of a particular PF, such that it transforms $G_0(x)$ to $G_{\mathcal{T}}(x)$ in a stable and efficient manner.

## 4.2 KEYPOINT FEATURE CALIBRATION

To reconstruct 3D keypoint (Liu et al., 2023) from one-shot demonstration, we simply incorporate trainable parameters into 3D Gaussians and align them with ground truth reference. Specifically, we employ widely adopted multi-view diffusion models $\phi_\theta$ (e.g., Zero1-to-3) to generate an image sequence $\mathcal{V}^{SRef} = \{I_i^{SRef}\}_{i=1}^{M} \in \mathbb{R}^{M,C,H,W}$ conditioned on viewpoints $O = \{o_i\}_{i=1}^{M}$ and using $I^{sup}$ as the support image.

$$\mathcal{V}^{SRef} = \phi_\theta^{SRef}(z; I^{sup}, O)$$

In order to leverage keypoint guidance, we introduce new trainable parameters $\mathcal{P}$ for 3D Gaussian Splatting, enabling keypoint supervision:

$$\mathcal{GS}(\mathcal{X}, \mathcal{C}, \alpha, r, s, \mathcal{P})$$

Here, $\mathcal{X} \in \mathbb{R}^{N,3}$ denotes the position, $\mathcal{C} \in \mathbb{R}^{N,H}$ represents the color, $\alpha \in \mathbb{R}^N$ is the opacity, $r \in \mathbb{R}^{N,3,3}$ is the rotation factor, $s \in \mathbb{R}^N$ is the scale factor, and $\mathcal{P} \in \mathbb{R}^{N,J}$ represents the keypoint parameters, where $J$ is the number of keypoints in a single image. Following equation 1, the predicted keypoints $\mathcal{K}^{SPred}$ are rendered through 3D GS as follows:

$$\mathcal{M}(x, y) = \sum_{i=1}^{n} T_i(x, y)\beta_i(x, y)\mathcal{P}_i(x, y) \tag{7}$$

$$\mathcal{K}^{SPred} = \text{concat}[\text{softmax}(\mathcal{M}(\cdot, y)), \text{softmax}(\mathcal{M}(x, \cdot))] \tag{8}$$

Here, $\mathcal{M}(x, y)$ is the keypoint estimation logit projected from 3D GS to the 2D image space $(x, y)$. Thus $\mathcal{M} \in \mathbb{R}^{M,C,H,W}$ represents the keypoint heatmap, and $\mathcal{K}^{SPred} \in \mathbb{R}^{M,J,2}$ denotes the rendered keypoints. Next, thanks to the cutting-edge 2D pose keypoint prediction models (?) $\psi_\theta$ (e.g., PoseAnything), we obtain the 2D keypoints $\mathcal{K}^{Ref} = \{P_i^{Ref}\}_{i=1}^{M} \in \mathbb{R}^{M,J,2}$ for each frame $I_i^{Ref}$ derived from $I^{sup}$, $P^{sup}$, and $\mathcal{V}^{SRef}$.

$$\mathcal{K}^{SRef} = \psi_\theta^{SRef}(\mathcal{V}^{SRef}; I^{sup}, P^{sup})$$

To enforce the model to learn the optimal spatial PF for keypoints during 3D GS, we take Euclidean distance as potential $U(x, \tau)$ in equation 4. The Keypoint Feature Calibration (KFC) Loss is then formalized as follows:

$$\mathcal{L}_{KFC} = U(x, \tau) = \|\mathcal{K}^{SPred} - \mathcal{K}^{SRef}\|_2^2 \tag{9}$$

Figure 3: **Training objectives of KG4D.** To achieve keypoint supervision, we design two loss functions to ensure alignment of Pose and Motion with the reference. (a) We introduce Keypoint Feature Calibration (KFC), where the keypoint matching error on the 2D image plane supervises the 3D Gaussian Splatting process. (b) We implement a Wasserstein Gradient Flow-based method, which computes the Wasserstein offset $\mathcal{H}$ for 2D GMMs $\mathcal{O}_{p_t}, \mathcal{O}_{p'_t}$ and compares it to the ground truth optical flow $\mathcal{F}(p_t, o^{TRef})$ of keypoints $\mathcal{K}^{TRef}$. This gradient flow facilitates smoother learning of the deformation network for 4D Gaussian Splatting (Bahmani et al., 2024).

## 4.3 MOTION CAPTURING VIA WASSERSTEIN GRADIENT FLOW

In the temporal motion capturing phase, we aim to align the Gaussian offsets corresponding to keypoints with the ground truth optical flow of those keypoints for accurate motion reconstruction. Specifically, we first reconstruct the missing temporal dimension in the implicit 4D ground truth derived from the single images. Then we utilize advanced video generation models (*e.g.*, SVD) to generate a single-view video sequence, denoted as $\mathcal{V}^{TRef} = \{I_t^{TRef}\}_{t=1}^T \in \mathbb{R}^{T,C,H,W}$, based on the support image $I^{sup}$. This process is formalized as:

$$\mathcal{V}^{TRef} = \phi_\theta^{TRef}(z; I^{sup}),$$

where $\phi_\theta^{TRef}$ represents the video generation function parameterized by $\theta$. To further implement 2D motion guidance from $\mathcal{V}^{TRef}$, we calculate optical flow of each keypoint as ground truth of motion variance. Specifically, we first compute keypoints $\mathcal{K}_t^{TRef} \in \mathbb{R}^{T,J,2}$ similarly to equation 7 and 8. Then, we obtain optical flow $\mathcal{F}_{t,t'}(p_t, o^{TRef}) \in \mathbb{R}^{J,2}$ of each pixel $p_t$ corresponding to $\mathcal{K}_t^{TRef}$ between consecutive $t$ and $t'$, where $t' > t$. While for the predicted motion variance of 3D GS, we propose Wasserstein offsets, which measure keypoint offsets in Wasserstein metric, leveraging optimal transport to enhance the optimization of the temporal PF. Specifically, we first project 3D Gaussians (Chen & Wang, 2024) onto the 2D image plane following equation 1:

$$\hat{G}(x; \hat{\mu}, \hat{\Sigma}) = f(G(x; \mu, \Sigma), o^{TRef})$$

where $\hat{\mu} = Q\mu, \hat{\Sigma} = Q\Sigma Q^T, Q \in \mathbb{R}^{2,3}$ denotes projection matrix, and $o^{TRef}$ represents the original camera pose derived from $I^{sup}$. After projecting the 3D Gaussian onto the 2D space, for each pixel $p_t$ corresponding to $\mathcal{K}_{t,i}^{Pred}$ at time $t$, where $i \in \{1, \ldots, J\}$, we compute the set of 2D Gaussians $\{\hat{G}(x)\}_{k=1}^K$ that contribute to the rendering process of $p_t$, and further reparameterize them into 2D GMMs:

$$\mathcal{O}_{p_t}(x) = \sum_{k=1}^K \tilde{\alpha}_k \hat{G}_k(x; \hat{\mu}_k, \hat{\Sigma}_k), \quad \tilde{\alpha}_k = \frac{\exp(\alpha_k)}{\sum_{k=1}^K \exp(\alpha_k)}$$

To model the temporal dimension with a better motion consistency, our solution is to constrain PFs of 4D Gaussians to follow the motion trajectories given by 2D optical flows. To achieve this, we propose Wasserstein offsets to represent offsets between two adjacent 2D GMMs $\mathcal{O}_{p_t}$ and $\mathcal{O}_{p'_t}$. Specifically, we calculate Wasserstein offset in Wasserstein metric, thus firstly obtain Wasserstein distance between Gaussian components as:

$$D_{k,k'} = \mathcal{W}_2^2(\hat{G}_{k,t}, \hat{G}_{k',t'}) = \|\hat{\mu}_{k,t} - \hat{\mu}_{k',t'}\|_2^2 + \text{Tr}\left(\hat{\Sigma}_{k,t} + \hat{\Sigma}_{k',t'} - 2\left(\hat{\Sigma}_{k,t}^{1/2}\hat{\Sigma}_{k',t'}\hat{\Sigma}_{k,t}^{1/2}\right)^{1/2}\right)$$

Next, we define the optimal transport plan $\gamma_{k,k'}$ to measure the mass transport from Gaussian component $\hat{G}_{k,1}$ to $\hat{G}_{k',2}$. To solve for $\gamma_{k,k'}$, we apply the Sinkhorn algorithm with entropy regularization. By iteratively updating the transport plan $\gamma_{k,k'}$ via the Sinkhorn algorithm[1], we efficiently compute the solution. Once the optimal transport plan is determined, the total Wasserstein distance is given by:

$$\mathcal{W}^2(\mathcal{O}_{p_t}, \mathcal{O}_{p_{t'}}) = \sum_{k=1}^{K} \sum_{k'=1}^{K} \gamma_{k,k'} D_{k,k'}$$

For simplification, we assume that each Gaussian has a scalar covariance matrix, *i.e.*, $S = \sigma I$ and $\Sigma = \sigma^2 I$. In this case, we define the Wasserstein offset between the 2D GMMs as:

$$\mathcal{H}(\mathcal{O}_{p_t}, \mathcal{O}_{p_{t'}}) = \sum_{k=1}^{K} \sum_{k'=1}^{K} \gamma_{k,k'} (\hat{\mu}_{k',t'} - \hat{\mu}_{k,t}) \tag{10}$$

Finally, we can optimize Wasserstein Gradient Flow as temporal PF through gradient learning of 3D GS:

$$\mathcal{L}_{WGF} = U(x, \tau) = \|\mathcal{F}_{t,t'}(p_t, o^{TRef}) - \mathcal{H}(\mathcal{O}_{p_t}, \mathcal{O}_{p_{t'}})\|_2^2 \tag{11}$$

## 5 EXPERIMENT

### 5.1 EXPERIMENT OVERVIEW

We comprehensively evaluated the performance of the proposed KG4D model in dynamic 4D scene generation using two approaches: from static images (randomly selecting 8 images from the DG4D and Animate124 datasets) and from videos (utilizing the Consistent4D dataset and challenging videos collected online). We used 14 key points as 3D Gaussian distribution parameters (set to 100, with the 4D model inheriting the 3D parameter settings) and incorporated Keypoint Feature Calibration Loss (KFC Loss) and Wasserstein Gradient Flow Loss (WGF Loss). All experiments were conducted on a 24GB 4090 GPU.

### 5.2 EXPERIMENTAL SETUP AND EVALUATION METRICS

This study employs a phased camera pose training and Gaussian distribution strategy. In the first phase, the focus is on pose optimization (using Keypoint Feature Calibration loss (KFC = 100) as the key weight) with 500 iterations; in the second phase, feature fine-tuning is conducted (50 iterations) without training geometry. The camera radius is set to 2, and the field of view is 49.1 degrees. Gaussian distribution sampling involves 5,000 points with an initial density of 10%, which is gradually increased over 3,000 iterations, dynamically optimizing position and opacity. The learning rate for the deformation field is maintained constant at 0.00064 and is slowly initiated using a delay multiplier. The learning rate for the grid is set to 0.0064, and Harmonic Spatio-temporal Encoding ($\omega_k = 1$) is used for encoding.

In this experiment, we utilize the DG4D and Animate124 datasets for the task of generating 4D scenes from static images, randomly selecting five static images for testing. Additionally, our method also supports the generation of 4D scenes from videos. For video inputs, we perform quantitative evaluations using the Consistent4D dataset. To comprehensively assess the quality of the generated results, we employ a variety of evaluation metrics, including LPIPS, FVD, PSNR, SSIM, FID, and FV4D. Specifically, LPIPS measures the perceptual differences between generated images and real images, FVD evaluates the quality of generated videos, PSNR is used to measure the peak signal-to-noise ratio of images, SSIM assesses the structural similarity of images, FID measures the distribution differences between generated and real images, and FV4D is dedicated to evaluating the quality of 4D scene generation. These metrics provide a comprehensive evaluation of the generated results from multiple dimensions, including perceptual quality, structural similarity, signal-to-noise ratio, and distribution consistency.

---

[1]Details are presented at Appendix. B

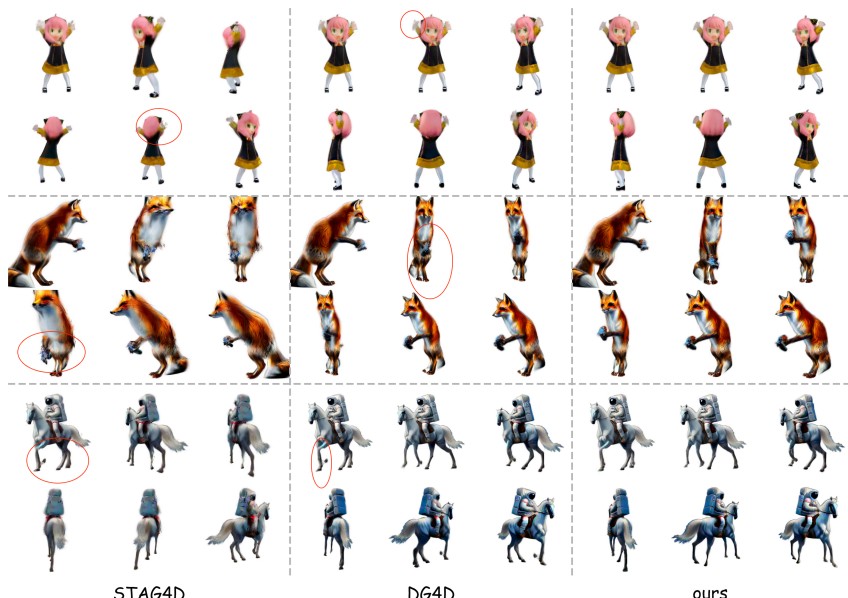

Figure 4: Generated 4D content of the comparative experiment.

## 5.3 EXPERIMENTAL COMPARISON WITH STATE-OF-THE-ART IMAGE-TO-4D METHODS

In this section, we experimentally compare our model with several state-of-the-art Image-to-4D methods, including representative works such as DG4D, Stag4D (Zeng et al., 2024), and Animate124. Through quantitative analysis across multiple metrics under the same experimental setup, we comprehensively evaluate the performance of each method in the task of generating 4D scenes. These methods represent various technical approaches in the field, covering multiple implementations for generating 4D scenes from static images.

| Model | FVD↓ | LPIPS↑ | PSNR↑ | SSIM↑ | FID↓ | FV↓ |
|-------|------|--------|-------|-------|------|-----|
| STAG4D | 657.94 | 0.0498 | 16.77 | 0.7046 | 0.0764 | 0.0764 |
| DG4D | 143.68 | 0.0798 | 26.77 | 0.9046 | 0.0064 | 0.0064 |
| **Ours** | **127.95** | **0.0908** | **29.64** | **0.9087** | **0.0057** | **0.0062** |

Table 1: Quantitative result of comparison across different models.

## 5.4 EXPERIMENTAL COMPARISON WITH VIDEO-TO-4D METHODS

In this section, we compare our model with several state-of-the-art Video-to-4D methods, including representative approaches such as SC4D (Wu et al., 2024), DG4D, SV4D (Xie et al., 2024), Consistent4D, and Stag4D. Under the same experimental setup, we quantitatively evaluate the generated results of these methods, using multiple evaluation metrics to comprehensively assess the performance of each method in the task of generating 4D scenes from video input. These methods represent different technical approaches for generating 4D scenes from video input, covering various mainstream methodologies in the field.

## 5.5 ABLATION STUDY

To further investigate the contribution of each component of the KG4D model to the overall performance, we designed the following ablation experiments: removing the Wasserstein Gradient Flow Loss (WGF Loss), removing both the Keypoint Feature Calibration Loss (KFC Loss) and WGF Loss, and evaluating the impact of different numbers of keypoints on model performance.

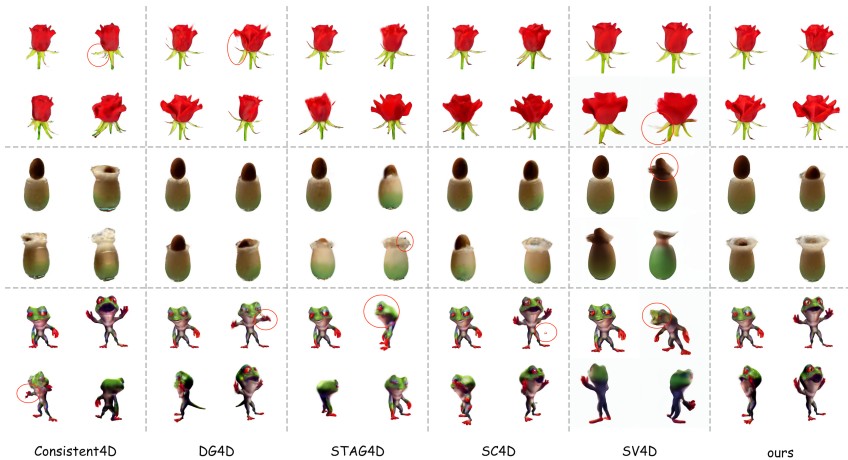

Figure 5: Visualization result of ablation study.

| Model | FVD↓ | LPIPS↑ | PSNR↑ | SSIM↑ | FID↓ |
|---|---|---|---|---|---|
| W/O(WGF) | 275.16 | 0.195 | 22.71 | 0.5946 | 0.0884 |
| W/O(WGF&KFC) | 463.49 | 0.218 | 52.77 | 0.3496 | 0.1359 |
| **Ours** | **226** | 0.128 | 29.14 | **0.8917** | **0.0059** |

Table 2: Quantitative evaluation of ablation study

Figure 6: Quantitative result of comparison across different models.

All ablation experiments were conducted under the same experimental setup to ensure fairness and comparability of the results.

## 6 DISCUSSION

In this work, we introduce KG4D, a framework that significantly advances 4D scene generation by effectively capturing spatio-temporal dynamics. Our Harmonic Spatio-temporal Encoding (HSE) and Keypoint Feature Calibration (KFC) ensure precise alignment and motion consistency, achieving state-of-the-art results in dynamic scene rendering. Future research could explore enhancing model efficiency for real-time applications, integrating advanced architectures, and expanding to more complex scenes. Additionally, incorporating other modalities like audio could lead to more immersive experiences. By refining KG4D, we aim to further bridge static inputs and dynamic outputs in neural rendering.

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

## A    Theorem Proofs

**Lemma 1** *Given two Gaussian mixture distributions $P_0$ and $P_1$ in $\mathbb{R}^4$, there exists a continuous probability flow $\varphi_\tau$, $\tau \in [0,1]$, such that $\varphi_0 = P_0$ and $\varphi_1 = P_1$. This flow can be described by a time-dependent probability density function $p(x, \tau)$, governed by equation 3.*

**Theorem 1** *For a Gaussian mixture distribution in $N$-dimensional space, the total probability flow $\varphi_t(x_1, \ldots, x_N)$ can be decomposed as $\varphi_t(x_1, \ldots, x_N) = \sum_{i=1}^{N} \mathbf{e}_i \varphi_t^{(i)}(x_i)$, where each $\varphi_t^{(i)}(x_i)$ represents the flow in the $i$-th dimension and satisfies equation 3.*

**Proof 1** *Consider a Gaussian mixture distribution in $N$-dimensional space with the probability density function (PDF) given by:*

$$\rho(\boldsymbol{x}, t) = \sum_k w_k \, G_k(\boldsymbol{x}),$$

*We assume the same as Section 4.3 that the dimensions are mutually independent, implying that the covariance matrix of each Gaussian component is diagonal. Therefore, each $G_k(\boldsymbol{x})$ can be expressed as a product of its marginal densities:*

$$G_k(\boldsymbol{x}) = \prod_{i=1}^{N} G_k^{(i)}(x_i),$$

*where $G_k^{(i)}(x_i)$ is the marginal Gaussian density of the $i$-th dimension for the $k$-th component.*

*The total probability density function becomes:*

$$\rho(\boldsymbol{x}, t) = \prod_{i=1}^{N} \rho^{(i)}(x_i, t),$$

*where:*

$$\rho^{(i)}(x_i, t) = \sum_k w_k \, G_k^{(i)}(x_i)$$

*is the marginal probability density in the $i$-th dimension.*

*The probability flow in $N$-dimensional space is defined as:*

$$\varphi_t(\boldsymbol{x}) = \rho(\boldsymbol{x}, t)\, \mathbf{v}(\boldsymbol{x}, t),$$

*where $\mathbf{v}(\boldsymbol{x}, t) = (v^{(1)}(x_1, t), v^{(2)}(x_2, t), \ldots, v^{(N)}(x_N, t))$ is the velocity field, and each $v^{(i)}(x_i, t)$ depends only on $x_i$ due to independence.*

*Because the dimensions are independent, the total probability flow vector can be expressed component-wise:*

$$\varphi_t(\boldsymbol{x}) = \Big(\rho(\boldsymbol{x}, t)\, v^{(1)}(x_1, t), \rho(\boldsymbol{x}, t)\, v^{(2)}(x_2, t), \ldots, \rho(\boldsymbol{x}, t)\, v^{(N)}(x_N, t)\Big).$$

*We define the flow in the $i$-th dimension as:*

$$\varphi_t^{(i)}(x_i) = \rho(\boldsymbol{x}, t)\, v^{(i)}(x_i, t).$$

*However, since $\rho(\boldsymbol{x}, t) = \rho^{(i)}(x_i, t)\, \rho^{(-i)}(\boldsymbol{x}_{-i}, t)$, where $\rho^{(-i)}(\boldsymbol{x}_{-i}, t) = \prod_{j \neq i} \rho^{(j)}(x_j, t)$, we can write:*

$$\varphi_t^{(i)}(x_i) = \rho^{(i)}(x_i, t)\, \rho^{(-i)}(\boldsymbol{x}_{-i}, t)\, v^{(i)}(x_i, t).$$

*Using the product rule for differentiation, the time derivative of $\rho(\boldsymbol{x}, t)$ is:*

$$\frac{\partial \rho(\boldsymbol{x}, t)}{\partial t} = \sum_{i=1}^{N} \left( \frac{\partial \rho^{(i)}(x_i, t)}{\partial t} \prod_{j \neq i} \rho^{(j)}(x_j, t) \right).$$

*Since each $\varphi_t^{(i)}(x_i)$ depends only on $x_i$, the divergence of $\varphi_t(\boldsymbol{x})$ is:*

$$\nabla \cdot \varphi_t(\boldsymbol{x}) = \sum_{i=1}^{N} \frac{\partial \varphi_t^{(i)}(x_i)}{\partial x_i}.$$

*Substituting $\varphi_t^{(i)}(x_i) = \rho(\boldsymbol{x}, t)\, v^{(i)}(x_i, t)$, we have:*

$$\frac{\partial \varphi_t^{(i)}(x_i)}{\partial x_i} = \left( \frac{\partial}{\partial x_i} \left[ \rho^{(i)}(x_i, t)\, v^{(i)}(x_i, t) \right] \right) \prod_{j \neq i} \rho^{(j)}(x_j, t).$$

*Follows equation 3, the continuity equation becomes:*

$$\frac{\partial \rho(\boldsymbol{x}, t)}{\partial t} + \nabla \cdot \varphi_t(\boldsymbol{x}) = \sum_{i=1}^{N} \left( \frac{\partial \rho^{(i)}(x_i, t)}{\partial t} \prod_{j \neq i} \rho^{(j)}(x_j, t) \right) + \sum_{i=1}^{N} \left( \frac{\partial \varphi_t^{(i)}(x_i)}{\partial x_i} \right)$$

$$= \sum_{i=1}^{N} \left( \left[ \frac{\partial \rho^{(i)}(x_i, t)}{\partial t} + \frac{\partial}{\partial x_i} \left( \rho^{(i)}(x_i, t)\, v^{(i)}(x_i, t) \right) \right] \prod_{j \neq i} \rho^{(j)}(x_j, t) \right)$$

$$= 0.$$

*Since $\prod_{j \neq i} \rho^{(j)}(x_j, t) > 0$, the expression inside the brackets must be zero for each $i$:*

$$\frac{\partial \rho^{(i)}(x_i, t)}{\partial t} + \frac{\partial}{\partial x_i} \left( \rho^{(i)}(x_i, t)\, v^{(i)}(x_i, t) \right) = 0.$$

*This is precisely the continuity equation for the $i$-th dimension.*

*Therefore, the total probability flow $\varphi_t(\boldsymbol{x})$ can be decomposed into the sum of the flows in each dimension:*

$$\varphi_t(\boldsymbol{x}) = \sum_{i=1}^{N} \mathbf{e}_i\, \varphi_t^{(i)}(x_i),$$

*where $\varphi_t^{(i)}(x_i) = \rho(\boldsymbol{x}, t) \, v^{(i)}(x_i, t)$.*

*Each flow $\varphi_t^{(i)}(x_i)$ satisfies the continuity equation in its respective dimension:*

$$\frac{\partial \rho^{(i)}(x_i, t)}{\partial t} + \frac{\partial \varphi_t^{(i)}(x_i)}{\partial x_i} = 0.$$

**Theorem 2** *Consider a one-dimensional isotropic Gaussian probability density function (PDF) $G(x, t; \mu(t), \sigma(t))$ with mean $\mu(t)$ and standard deviation $\sigma(t)$, where both $\mu(t)$ and $\sigma(t)$ are time-dependent parameters. Define the joint loss function as:*

$$\mathcal{L}(\mu, \sigma) = \frac{1}{2}(\mu(t) - \mu^*)^2 + \frac{1}{2}(\sigma(t) - \sigma^*)^2, \tag{12}$$

*where $\mu^*$ and $\sigma^*$ are target values for the mean and standard deviation, respectively. Suppose the parameters $\mu(t)$ and $\sigma(t)$ evolve according to the gradient descent updates:*

$$\frac{d\mu}{dt} = -\nabla_\mu \mathcal{L} = \mu^* - \mu(t),$$

$$\frac{d\sigma}{dt} = -\nabla_\sigma \mathcal{L} = \sigma^* - \sigma(t).$$

*Define the probability flow $\varphi_t(x)$ as:*

$$\varphi_t(x) = (\mu^* - \mu(t)) + \frac{\sigma^* - \sigma(t)}{\sigma(t)}(x - \mu(t)).$$

*Then, the probability density function $G(x, t; \mu(t), \sigma(t))$ satisfies the continuity equation:*

$$\frac{\partial G}{\partial t} + \frac{\partial}{\partial x}(\varphi_t(x)G) = 0,$$

*and the energy functional $E(\varphi_t)$ defined by:*

$$E(\varphi_t) = \int_{\mathbb{R}} \frac{1}{2}\varphi_t(x)^2 G(x, t; \mu(t), \sigma(t)) \, dx,$$

*is exactly equal to the loss function:*

$$E(\varphi_t) = \mathcal{L}(\mu(t), \sigma(t)).$$

**Proof 2** *We will demonstrate the theorem in two main parts: (1) Verification of the continuity equation 3 to confirm the process of gradient descent of loss function 12 is equivalent to a probability flow. (2) Establishment of the equivalence between the energy function of the flow and the loss function.*

### *Part 1: Verification of the Continuity Equation*

*The one-dimensional Gaussian distribution is given by:*

$$G(x, t; \mu(t), \sigma(t)) = \frac{1}{\sqrt{2\pi\sigma(t)^2}} \exp\left(-\frac{(x - \mu(t))^2}{2\sigma(t)^2}\right),$$

*where $\mu(t)$ and $\sigma(t)$ are time-dependent parameters governing the mean and standard deviation of the distribution, respectively.*

*Using the chain rule, the time derivative of $G$ is:*

$$\frac{\partial G}{\partial t} = \frac{d\mu}{dt} \cdot \frac{\partial G}{\partial \mu} + \frac{d\sigma}{dt} \cdot \frac{\partial G}{\partial \sigma}.$$

*The partial derivatives are given by:*

$$\frac{\partial G}{\partial \mu} = \frac{(x - \mu(t))}{\sigma(t)^2}G,$$

$$\frac{\partial G}{\partial \sigma} = \left( -\frac{1}{\sigma(t)} + \frac{(x - \mu(t))^2}{\sigma(t)^3} \right) G.$$

*Substituting the gradient descent updates:*

$$\frac{d\mu}{dt} = \mu^* - \mu(t),$$

$$\frac{d\sigma}{dt} = \sigma^* - \sigma(t),$$

*we obtain:*

$$\frac{\partial G}{\partial t} = (\mu^* - \mu(t)) \cdot \frac{(x - \mu(t))}{\sigma(t)^2} G + (\sigma^* - \sigma(t)) \left( -\frac{1}{\sigma(t)} + \frac{(x - \mu(t))^2}{\sigma(t)^3} \right) G.$$

*Define the probability flow $\varphi_t(x)$ as:*

$$\varphi_t(x) = (\mu^* - \mu(t)) + \frac{\sigma^* - \sigma(t)}{\sigma(t)} (x - \mu(t)).$$

*The divergence of the product $\varphi_t(x)G$ is:*

$$\frac{\partial}{\partial x}(\varphi_t(x)G) = \frac{\partial}{\partial x}(\varphi_t(x)) G + \varphi_t(x) \frac{\partial G}{\partial x}.$$

*First, compute $\frac{\partial \varphi_t(x)}{\partial x}$:*

$$\frac{\partial \varphi_t(x)}{\partial x} = \frac{\sigma^* - \sigma(t)}{\sigma(t)}.$$

*Next, compute $\frac{\partial G}{\partial x}$:*

$$\frac{\partial G}{\partial x} = -\frac{(x - \mu(t))}{\sigma(t)^2} G.$$

*Thus, the divergence becomes:*

$$\frac{\partial}{\partial x}(\varphi_t(x)G) = \frac{\sigma^* - \sigma(t)}{\sigma(t)} G - \left( (\mu^* - \mu(t)) + \frac{\sigma^* - \sigma(t)}{\sigma(t)} (x - \mu(t)) \right) \frac{(x - \mu(t))}{\sigma(t)^2} G.$$

*Simplify the expression:*

$$\frac{\partial}{\partial x}(\varphi_t(x)G) = \frac{\sigma^* - \sigma(t)}{\sigma(t)} G - (\mu^* - \mu(t)) \frac{(x - \mu(t))}{\sigma(t)^2} G - \frac{\sigma^* - \sigma(t)}{\sigma(t)^3} (x - \mu(t))^2 G.$$

*The continuity equation requires:*

$$\frac{\partial G}{\partial t} + \frac{\partial}{\partial x}(\varphi_t(x)G) = 0.$$

*Substitute the expressions for $\frac{\partial G}{\partial t}$ and $\frac{\partial}{\partial x}(\varphi_t(x)G)$ and simplify the terms:*

$$(\sigma^* - \sigma(t)) \left[ \left( -\frac{1}{\sigma(t)} + \frac{(x - \mu(t))^2}{\sigma(t)^3} \right) + \frac{1}{\sigma(t)} - \frac{(x - \mu(t))^2}{\sigma(t)^3} \right] G = 0.$$

*The equation holds, and thus the continuity equation is satisfied.*

**Part 2: Equivalence Between the Energy Functional and the Loss Function**

*The energy functional is defined as:*

$$E(\varphi_t) = \int_{\mathbb{R}} \frac{1}{2} \varphi_t(x)^2 G(x, t; \mu(t), \sigma(t)) \, dx.$$

*Substitute $\varphi_t(x)$:*

$$\varphi_t(x) = (\mu^* - \mu(t)) + \frac{\sigma^* - \sigma(t)}{\sigma(t)} (x - \mu(t)).$$

*Thus, the energy functional becomes:*

$$E(\varphi_t) = \frac{1}{2} \int_{\mathbb{R}} \left[ (\mu^* - \mu(t)) + \frac{\sigma^* - \sigma(t)}{\sigma(t)} (x - \mu(t)) \right]^2 G(x, t; \mu(t), \sigma(t)) \, dx.$$

*Since $x$ is distributed according to $G(x, t; \mu(t), \sigma(t))$, we use the following properties:*

$$\mathbb{E}[x - \mu(t)] = 0,$$

$$\mathbb{E}[(x - \mu(t))^2] = \sigma(t)^2.$$

*Thus, we compute each term in the expansion of $E(\varphi_t)$.*

*1. First Term:*

$$\int_{\mathbb{R}} (\mu^* - \mu(t))^2 G(x, t; \mu(t), \sigma(t)) \, dx = (\mu^* - \mu(t))^2.$$

*2. Second Term:*

$$2(\mu^* - \mu(t)) \frac{\sigma^* - \sigma(t)}{\sigma(t)} \int_{\mathbb{R}} (x - \mu(t)) G(x, t; \mu(t), \sigma(t)) \, dx = 0.$$

*3. Third Term:*

$$\left( \frac{\sigma^* - \sigma(t)}{\sigma(t)} \right)^2 \int_{\mathbb{R}} (x - \mu(t))^2 G(x, t; \mu(t), \sigma(t)) \, dx = (\sigma^* - \sigma(t))^2.$$

*Summing the results gives:*

$$E(\varphi_t) = \frac{1}{2} \left[ (\mu^* - \mu(t))^2 + (\sigma^* - \sigma(t))^2 \right] = \mathcal{L}(\mu(t), \sigma(t)).$$

**Theorem 3** *Let $GMM(t)$ denote a Gaussian Mixture Model at time $t$, which consists of a weighted sum of $K$ Gaussian components. The Wasserstein offset $\psi(\rho(t), \rho(t+1))$ between adjacent time steps is defined as:*

$$\psi(\rho(t), \rho(t+1)) = \sum_{k=1}^{K} \sum_{k'=1}^{K} \gamma_{k,k'} (\mu_{k',t+1} - \mu_{k,t})$$

*where $\mu_{k,t}$ is the mean of the $k$-th Gaussian component at time $t$, and $\gamma_{k,k'}$ is the transport matrix determined by solving the following optimal transport problem:*

$$\min_{\gamma_{k,k'}} \sum_{k=1}^{K} \sum_{k'=1}^{K} \gamma_{k,k'} \| \mu_{k,t} - \mu_{k',t+1} \|^2$$

*subject to the transport constraints:*

$$\sum_{k'} \gamma_{k,k'} = w_k(t), \quad \sum_{k} \gamma_{k,k'} = w_{k'}(t+1)$$

*where $w_k(t)$ and $w_{k'}(t+1)$ are the weights of the Gaussian components at time $t$ and $t+1$, respectively.*

*Let the energy functional be defined as:*

$$\mathcal{F}[\rho(t)] = \frac{1}{N} \sum_{i=1}^{N} \| \mathbf{y}_i(t) - \psi(\rho(t), \rho(t+1)) \|^2$$

*where $\mathbf{y}_i(t)$ represents the target offset. The velocity field of the probability flow $\mathbf{v}(\tau)$, driven by this energy functional, is the gradient of the functional's variational derivative. This probability flow satisfies the Wasserstein gradient flow (WGF).*

**Proof 3** *We aim to show that $\mathcal{F}[\rho(t)]$ satisfies the following conditions:*

*1. Initial Condition: The energy functional $\mathcal{F}$ must be finite at the initial measure $\rho_0$, i.e., $\mathcal{F}[\rho_0] < +\infty$.*

2. *Lower Semicontinuity: The energy functional $\mathcal{F}$ is lower semicontinuous in the weak topology.*

3. *Boundary Condition: The flow must satisfy the no-flux boundary condition (Neumann boundary condition), ensuring mass conservation.*

4. *Convexity: The energy functional $\mathcal{F}$ is $\lambda$-geodesically convex in the Wasserstein space.*

*If these conditions hold, the Fokker-Planck equation can be viewed as a gradient flow in the Wasserstein space (Santambrogio, 2017).*

### Part 1: Initial Condition

*We need to show that the energy functional $\mathcal{F}[\rho(t)]$ is finite at the initial measure $\rho_0$, i.e., $\mathcal{F}[\rho_0] < +\infty$.*

*The energy functional is given by:*

$$\mathcal{F}[\rho(t)] = \frac{1}{N} \sum_{i=1}^{N} \|\mathbf{y}_i(t) - \psi(\rho(t), \rho(t+1))\|^2$$

*The Wasserstein offset $\psi(\rho(t), \rho(t+1))$ is defined as:*

$$\psi(\rho(t), \rho(t+1)) = \sum_{k=1}^{K} \sum_{k'=1}^{K} \gamma_{k,k'}(\mu_{k',t+1} - \mu_{k,t})$$

*Since $\gamma_{k,k'}$ is obtained from the optimal transport problem, it satisfies the transport constraints:*

$$\sum_{k'} \gamma_{k,k'} = w_k(t), \quad \sum_{k} \gamma_{k,k'} = w_{k'}(t+1)$$

*where $w_k(t)$ and $w_{k'}(t+1)$ are finite weights. Thus, $\psi(\rho(t), \rho(t+1))$ is a finite vector. Since the target offset $\mathbf{y}_i(t)$ is also finite, each term in the sum $\|\mathbf{y}_i(t) - \psi(\rho(t), \rho(t+1))\|^2$ is finite, and therefore:*

$$\mathcal{F}[\rho(t)] < +\infty$$

*Thus, the initial condition is satisfied.*

### Part 2: Lower Semicontinuity

*We need to show that the energy functional $\mathcal{F}[\rho(t)]$ is lower semicontinuous in the weak topology.*

*The energy functional is defined as:*

$$\mathcal{F}[\rho(t)] = \frac{1}{N} \sum_{i=1}^{N} \|\mathbf{y}_i(t) - \psi(\rho(t), \rho(t+1))\|^2$$

*The offset $\psi(\rho(t), \rho(t+1))$ is derived from the transport matrix $\gamma_{k,k'}$, which is obtained by solving an optimal transport problem in the weak topology. The optimal transport problem is continuous in this topology.*

*Since the norm squared $\|\cdot\|^2$ is a lower semicontinuous function, and the sum of lower semicontinuous functions is also lower semicontinuous, it follows that $\mathcal{F}[\rho(t)]$ is lower semicontinuous:*

$$\liminf_{n \to \infty} \mathcal{F}[\rho_n(t)] \geq \mathcal{F}[\rho(t)]$$

*Thus, the lower semicontinuity condition is satisfied.*

### Part 3: Boundary Condition

*We need to verify that the probability flow satisfies the no-flux boundary condition (Neumann boundary condition), ensuring mass conservation.*

*In the Wasserstein gradient flow framework, the boundary condition is implicitly satisfied by the transport problem. Specifically, the transport matrix $\gamma_{k,k'}$ satisfies the mass conservation constraints:*

$$\sum_{k'} \gamma_{k,k'} = w_k(t), \quad \sum_{k} \gamma_{k,k'} = w_{k'}(t+1)$$

*This ensures that the mass of each Gaussian component is conserved during the transport from $\rho(t)$ to $\rho(t+1)$. Therefore, the no-flux boundary condition is naturally satisfied, and mass is conserved throughout the evolution of the system.*

***Part 4: Convexity***

*We need to show that the energy functional $\mathcal{F}[\rho(t)]$ is $\lambda$-geodesically convex in the Wasserstein space.*

*The energy functional is given by:*

$$\mathcal{F}[\rho(t)] = \frac{1}{N} \sum_{i=1}^{N} \|\mathbf{y}_i(t) - \psi(\rho(t), \rho(t+1))\|^2$$

*Since the offset $\psi(\rho(t), \rho(t+1))$ is a linear combination of the transport matrix elements $\gamma_{k,k'}$, which are obtained by solving a linear minimization problem, and the squared norm $\| \cdot \|^2$ is a convex function, $\mathcal{F}[\rho(t)]$ is convex.*

*Furthermore, in the Wasserstein space, the squared distance function is geodesically convex. This implies that $\mathcal{F}[\rho(t)]$ is $\lambda$-geodesically convex in the Wasserstein space, i.e., for any two measures $\rho_0$ and $\rho_1$, and for any $s \in [0, 1]$, the following inequality holds:*

$$\mathcal{F}[\rho_s] \leq (1 - s)\mathcal{F}[\rho_0] + s\mathcal{F}[\rho_1]$$

*where $\rho_s$ is the geodesic between $\rho_0$ and $\rho_1$ in the Wasserstein space. Thus, the convexity condition is satisfied.*

## B  OPTIMAL TRANSPORT

Given two probability distributions, Optimal transport (OT) seeks to minimize the total transportation cost between them, where the cost is defined by a specified metric, often referred to as the ground cost. In modern machine learning, OT has found wide applications in domains such as computer vision, natural language processing, and generative modeling due to its ability to measure the distance between probability measures, aligning samples or features from different distributions.

Formally, Let $\mu$ and $\nu$ be two probability measures on measurable spaces $\mathcal{X}$ and $\mathcal{Y}$, respectively, and let $c(x, y)$ represent the cost of transporting a unit mass from point $x \in \mathcal{X}$ to point $y \in \mathcal{Y}$. The goal of OT is to find a joint probability distribution $\gamma \in \Pi(\mu, \nu)$, where $\Pi(\mu, \nu)$ denotes the set of all couplings of $\mu$ and $\nu$, that minimizes the total transportation cost:

$$\text{OT}(\mu, \nu) = \inf_{\gamma \in \Pi(\mu, \nu)} \int_{X \times Y} c(x, y) \, d\gamma(x, y)$$

In practice, solving this optimization problem directly is computationally expensive, especially for high-dimensional data, as it often leads to intractable computations. To address this, an entropy-regularized version of the OT problem, known as the Sinkhorn distance or Wasserstein distance, is often employed.

The Sinkhorn algorithm introduces an entropic regularization term to the OT formulation, making the optimization more tractable. The regularized OT problem is defined as follows:

$$\text{OT}_\epsilon(\mu, \nu) = \inf_{\gamma \in \Pi(\mu, \nu)} \left( \int_{\mathcal{X} \times \mathcal{Y}} c(x, y) \, d\gamma(x, y) + \epsilon H(\gamma) \right)$$

where $H(\gamma) = \int_{\mathcal{X} \times \mathcal{Y}} \gamma(x, y) \log \gamma(x, y) \, dx \, dy$ is the entropy of the coupling $\gamma$, and $\epsilon > 0$ is the regularization parameter. The regularization smooths the optimization landscape, allowing efficient computation via iterative scaling algorithms.

The Sinkhorn algorithm alternates between updating the row and column marginals of the coupling matrix using iterative scaling. It converges to a near-optimal solution while maintaining computational efficiency, making it suitable for large-scale applications.

**Algorithm 1** Sinkhorn Iteration for Optimal Transport

---

**Require:** Cost matrix $C \in \mathbb{R}^{n \times m}$, distributions $\mu \in \mathbb{R}^n$, $\nu \in \mathbb{R}^m$, regularization parameter $\epsilon > 0$, tolerance $\delta > 0$

**Ensure:** Approximate optimal transport plan $\gamma$

1: Initialize $u = \mathbf{1}_n$, $v = \mathbf{1}_m$
2: Compute kernel $K = \exp\left(-\frac{C}{\epsilon}\right)$
3: **while** not converged **do**
4:     $u \leftarrow \frac{\mu}{Kv}$                                                       $\triangleright$ Update row scaling
5:     $v \leftarrow \frac{\nu}{K^\top u}$                                                    $\triangleright$ Update column scaling
6:     **if** change in $u$ and $v$ is less than $\delta$ **then**
7:         **break**
8:     **end if**
9: **end while**
10: Compute transport plan $\gamma = \text{diag}(u)K\text{diag}(v)$
11: **return** $\gamma$

---

