# OpenReview forum: "KEYPOINT-GUIDED 4D GAUSSIAN SPLATTING WITH DECOUPLED SPATIO-TEMPORAL FLOW REFINEMENT"
_ICLR.cc/2025/Conference — ICLR 2025 Conference Withdrawn Submission_

### Official Review · Reviewer_AQWD · 2024-10-21

**Soundness:** 2
**Presentation:** 1
**Contribution:** 2
**Rating:** 3
**Confidence:** 4

**Summary:**

The paper presents a pipeline to generate 4D scene with a single image as input. A multi-view diffusion model is used to first generate a static 3D scene, and a video generation model is used to generate the motion as the forth dimension. Key point feature is used in both stages to enhance the consistency during the reconstruction.

**Strengths:**

1. The proposed idea is reasonable. Generating the spatial and temporal dimensions separately and using key points to avoid inconsistencies seems helpful.

**Weaknesses:**

The paper seems to have been written in a hurry and is not yet finished. The idea is sound, so I hope the author better prepare the paper before submission next time. For instance:

1. There are no citation referred in the first two sub-section of the related work (lines 91-107).
2. The abstraction mentioned "evaluation on various benchmarks" (line 32), but only on dataset is evaluated.
3. The experiment section mentioned the comparisons are made on various SOTA methods (lines 403-404, 420-421). However, only a few of them are evaluated.
4. The experiment is not convincing. The dataset is not clearly claimed in the Table title. The numbers are not match between the "ours" row in Table 1 and 2 (probably they are for different datasets, but I didn't see where is mentioned).
5. The experiment section 5.4 contains no result, all the Tables and image results are not referred and discussed.
6. The equations are hard to follow especially for the section 4.2. The letter "z" in the equation in line 243 is not defined. $K^{ref},P^{ref},I^{ref}$ in line 262 are not defined. A citation reference is missing in line 260.

**Questions:**

If the author would like to rebuttal, I do have several questions regarding to the proposed method.

1. As I know, the original DG4D dataset dose not contain key point ground truth, but the paper mentioned the ground truth keypoints $P^{sup}$ in line 215. Is this keypoint manually labelled by the author? Or how the ground truth keypoints are selected.
2. Is there a reason why different sets of keypoint are used for the spatial and temporal generation stages.

---

### Official Review · Reviewer_kF4Q · 2024-10-27

**Soundness:** 2
**Presentation:** 2
**Contribution:** 3
**Rating:** 5
**Confidence:** 3

**Summary:**

This paper proposes a novel method for generating time-aware 4D representations from a single static image or video. The proposed HSE  more effectively captures dynamic scene changes by decoupling spatial and temporal information. The KFC provides additional supervision of local patterns in spatial dimensions to ensure accurate keypoint alignment in 3D Gaussian Splatting. The WGF  is implemented to enhance motion consistency and stability, effectively reducing artifacts and improving visual coherence. Through comprehensive evaluations, this paper establishes a new benchmark for 4D Gaussian splatting-based methods, demonstrating KG4D's state-of-the-art performance in generating realistic, temporally consistent dynamic scenes.

**Strengths:**

1. The emphasis on decoupling spatial and temporal dimensions, along with the use of keypoint guidance, is a significant contribution.
2. The visualization results are impressive and effectively demonstrate the method's capabilities.

**Weaknesses:**

1. The writing requires improvement, particularly in the Method section.
   (a) The citation in line 147 is invalid.
   (b) The equation in line 257 is difficult to understand.
   (c) What does C refer to in line 259? Does it represent J?
   (d) There is a missing citation in line 260.
   (e) The description in section 4.3 is vague, especially from lines 324 to 341.

2. Regarding contributions:
   (a) How is the spatial and temporal components decoupled? A detailed description is lacking.
   (b) Will the design of keypoint feature calibration introduce a large number of parameters ($R^{N,J} $)? Could a self-supervised approach, similar to that used in SCGS[1], be better?
   (c) The section on motion capturing via wasserstein gradient flow needs more explanation to clarify the meaning of optimal transport.

3. Regarding experiments: The experimental section is somewhat lacking, with insufficient ablation studies. For example, the specific role of spatial-temporal decoupling and the visualization of keypoint feature calibration need to be better explained.
4. The authors did not provide video results.

[1] SC-GS: Sparse-Controlled Gaussian Splatting for Editable Dynamic Scenes

**Questions:**

See weaknesses above.

---

### Official Review · Reviewer_ZqC9 · 2024-11-01

**Soundness:** 1
**Presentation:** 1
**Contribution:** 2
**Rating:** 3
**Confidence:** 5

**Summary:**

This paper claims that they introduce a novel 4D generation framework, KG4D, which is featured by HSE, KFC, and WGF and outperforms existing state-of-the-art methods on various benchmarks in dynamic 4D generation and novel viewpoint synthesis.

**Strengths:**

The idea of endowing Gaussian with keypoint features and incorporating some regularization for them into generation is interesting. However, the solutions and results presented in this manuscripts are highly suspicious as they completely disregard basic academic standards obviously lacking the basic knowledge for this area.

**Weaknesses:**

The reviewer believes this manuscript should not deserve serious consideration as it is full filled with nonsense sounds totally generated by an AI. Both the motivation and the solutions lack precise and clear meaning, making it incompetent for an academic paper. Moreover, there are numerous errors throughout the whole paper, e.g. they even have not provide correct reference for 3D Gaussian Splatting.

**Questions:**

None

---

### Note · Authors · 2024-11-13

**Comment:**

Thank you all very much for the constructive and insightful comments of all the reviewers.

**Withdrawal Confirmation:**

I have read and agree with the venue's withdrawal policy on behalf of myself and my co-authors.